# STARS: Self-supervised Tuning for 3D Action Recognition in Skeleton Sequences

## Abstract

Self-supervised pretraining methods with masked prediction demonstrate remarkable within-dataset performance in skeleton-based action recognition. However, we show that, unlike contrastive learning approaches, they do not produce well-separated clusters. Additionally, these methods struggle with generalization in few-shot settings. To address these issues, we propose Self-supervised Tuning for 3D Action Recognition in Skeleton sequences (STARS). Specifically, STARS first uses a masked prediction stage using an encoder-decoder architecture. It then employs nearest-neighbor contrastive learning to partially tune the weights of the encoder, enhancing the formation of semantic clusters for different actions. By tuning the encoder for a few epochs, and without using hand-crafted data augmentations, STARS achieves state-of-the-art self-supervised results in various benchmarks, including NTU-60, NTU-120, and PKU-MMD. In addition, STARS exhibits significantly better results than masked prediction models in few-shot settings, where the model has not seen the actions throughout pretraining. Our code and trained weights are available at: `https://anonymous.4open.science/r/stars-CD2E`

## 1 Introduction

Human action recognition is receiving growing attention in computer vision due to its wide applications in the real world, such as security, human-machine interaction, medical assistance, and virtual reality Kazakos et al. (2019); Yang et al. (2019); Nikam & Ambekar (2016); Wei et al. (2014). While some previous works have focused on recognizing actions based on appearance information, other approaches have highlighted the benefits of using pose information. In comparison to RGB videos, Representing videos of human activities with 3D skeleton sequences offers advantages in privacy preservation, data efficiency, and excluding extraneous details such as background, lighting variations, or diverse clothing types. Recent models for 3D action recognition based on skeleton sequences have demonstrated impressive results Lee et al. (2023); Duan et al. (2022a;b); Chen et al. (2021). However, these models heavily depend on annotations, which are labor-intensive and time-consuming to acquire. Motivated by this, in this study, we investigate the self-supervised representation learning of 3D actions.

Prior studies in self-supervised learning have employed diverse pretext tasks, such as predicting motion and recognizing jigsaw puzzles Lin et al. (2020); Zheng et al. (2018); Su et al. (2020). More recently, current research has shifted its focus towards contrastive learning Lin et al. (2023); Mao et al. (2022; 2023b) or Mask Autoencoders (MAE) Wu et al. (2023); Mao et al. (2023a).

Contrastive learning approaches, although effective in learning representations, rely heavily on data augmentations to avoid focusing on spurious features. Without using data augmentations, they are prone to the problem of shortcut learning Geirhos et al. (2020), leading to potential overfitting on extraneous features, such as a person's height or the camera angle, which do not provide a valid cue to discriminate between different actions. As a result, some knowledge expert Tian et al. (2020) is needed to design different augmentations of the same sequence; and methods that incorporate extreme augmentations in their pretraining pipeline Guo et al. (2022); Lin et al. (2023) have shown significant improvements.

MAE-based methods mask out a proportion of the input sequence and use an encoder-decoder architecture to reconstruct the missing information from the input based on the latent representation of unmasked regions. These approaches Wu et al. (2023); Mao et al. (2023a) have recently outperformed their contrastive counterparts in within-dataset metrics. However, we show that representations learned by these models cannot separate different actions as effectively as contrastive learning-based methods in few-shot settings.

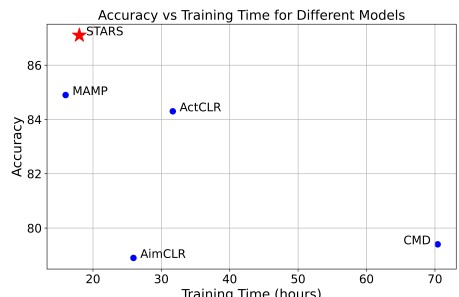

Figure 1: Comparison between training time and test-time accuracy on linear evaluation protocol. Training time is evaluated on a single NVIDIA GeForce RTX 3090 GPU.

Despite the significant efforts that have been made, how to learn a more discriminative representation of skeletons is still an issue for skeleton-based action recognition. We believe that integrating MAE-based approaches with contrastive-learning methods can enhance the generalizability of representations, while preserving the strong performance of MAE models in within-dataset evaluations. To this end, we propose Self-supervised Tuning for 3D Action Recognition in Skeleton Sequences (STARS), a simple and efficient self-supervised framework for 3D action representation learning. It is a sequential approach that initially uses MAE as the pretext task. In the subsequent stage, it trains a contrastive head in addition to partially tuning the encoder for a few epochs, motivating the representation to form distinct and well-separated clusters. Fig. 1 shows that STARS requires significantly less resources during pretraining compared to contrastive learning approaches. In addition, STARS outperforms both MAE and contrastive learning approaches.

In summary, our main contributions are as follows:

- We propose the STARS framework, a sequential approach that improves the MAE encoder output representation to create well-separated clusters, without any extra data augmentations, and with only a few epochs of contrastive tuning.

- We show that, although MAE approaches excel in within-dataset evaluations, they exhibit a lack of generalizability in few-shot settings. Subsequently, we significantly enhance their few-shot capabilities while maintaining their strong within-dataset performance by employing our method.

- We validate the efficacy of our approach through extensive experiments and ablations on three large datasets for 3D skeleton-based action recognition, attaining state-of-the-art performance in most cases.

## 2 RELATED WORK

### 2.1 SELF-SUPERVISED SKELETON-BASED ACTION RECOGNITION

The objective of self-supervised action recognition is to train an encoder to discriminate sequences with different actions without providing any labels throughout the training. Methods such as LongT-GAN Zheng et al. (2018) pretrain their model with 3D skeleton reconstruction using an encoder-decoder architecture; and P&C Su et al. (2020) improves the performance by employing a weak decoder. Colorization Yang et al. (2021) represents the sequence as 3D point clouds and colorizes it based on the temporal and spatial orders in the original sequence.

Several studies explored various contrastive learning approaches, showing promising results Li et al. (2021); Guo et al. (2022); Lin et al. (2023); Mao et al. (2022; 2023b). CrosSCLR Li et al. (2021) applies the MoCo He et al. (2020) framework and introduces cross-view contrastive learning. This approach aims to compel the model to maintain consistent decision-making across different views. AimCLR Guo et al. (2022) improves the representation by proposing extreme augmentations. CMD Mao et al. (2022) trains three encoders simultaneously and distills knowledge from one to another by introducing a new loss function. I$^2$MD Mao et al. (2023b) extends the CMD framework by introducing intra-modal mutual distillation, aiming to elevate its performance through incorpo-

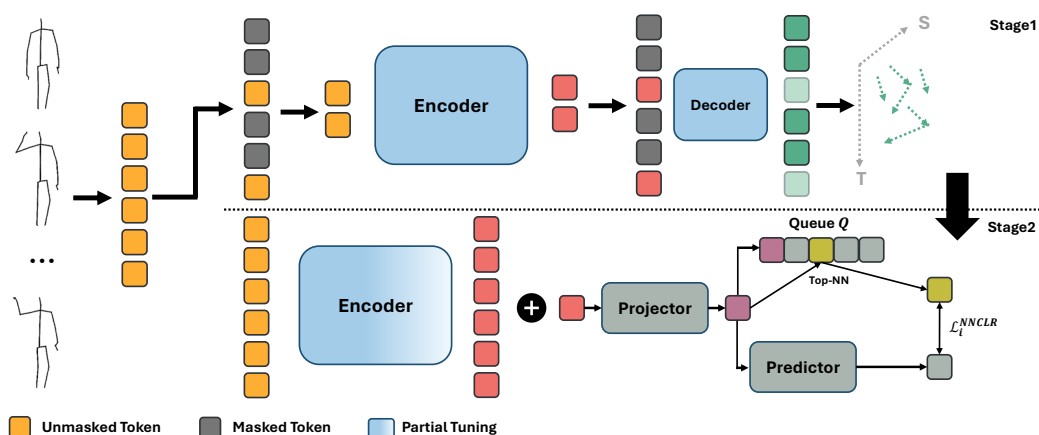

Figure 2: The overall pipeline of our proposed STARS framework. The first stage uses MAMP Mao et al. (2023a) to reconstruct the motion of masked tokens. The second stage trains parameters of the projector and predictor using a contrastive learning approach in addition to partially tuning the encoder weights.

rating local cluster-level contrasting. ActCLR Lin et al. (2023) employs an unsupervised approach to identify actionlets, which are specific body areas involved in performing actions. The method distinguishes between actionlet and non-actionlet regions and applies more severe augmentations to non-actionlet regions.

Recently, MAE-based approaches showed significant improvements. SkeletonMAE Wu et al. (2023) reconstructs the spatial positions of masked tokens. MAMP Mao et al. (2023a) uses temporal motion as its reconstruction target and proposes a motion-aware masking strategy. However, we show that MAE-based methods exhibit limited generalization in few-shot settings when compared to contrastive-learning based approaches.

### 2.2 COMBINING MASKED AUTOENCODERS WITH INSTANCE DISCRIMINATION

Some recent works in the image domain investigated the effect of combining MAE and Instance Discrimination (ID) methods Zhou et al. (2021); Wang et al. (2022); Mishra et al. (2022); Tao et al. (2023); Lehner et al. (2023). iBOT Zhou et al. (2021) combines DINO Caron et al. (2021) and BEiT Bao et al. (2021) for the pretext task. RePre Wang et al. (2022) extends the contrastive learning framework by adding pixel-level reconstruction loss. CAN Mishra et al. (2022) adds gaussian noise to the unmasked patches and it reconstructs the noise and masked patches, and adds a contrastive loss to the encoder output. MSN Assran et al. (2022) aligns an image view featuring randomly masked patches with the corresponding unmasked image. SiameseIM Tao et al. (2023) predicts dense representations from masked images in different views. MAE-CT Lehner et al. (2023) proposes a sequential training by adding contrastive loss after MAE training.

Our work is a sequential self-supervised approach for pretraining of skeleton sequences. It initially employs an MAE approach using an encoder-decoder architecture and further enhances the output representation of the encoder by tuning its weights using contrastive learning.

## 3 METHOD

### 3.1 FRAMEWORK OVERVIEW

The overall framework of STARS is illustrated in Fig. 2. It is a sequential self-supervised approach consisting of two main stages. The first stage relies on an MAE-like framework to pretrain the weights of the encoder. We use MAMP Mao et al. (2023a) because it shows promising result in 3D action representation learning; however, any alternative MAE-based approach is also applicable. The next stage is designed to tune the parameters of the encoder using an instance discrimination method. Specifically, the second stage replaces the decoder with a projector and predictor Grill et al. (2020). It trains them in addition to the encoder using Nearest-Neighbor Contrastive Learning (NNCLR) Dwibedi et al. (2021) to converge to a representation capable of discriminating different

sequences. This approach helps the encoder learn to output distinct clusters for different actions, improving its ability to discriminate between various sequences.

## 3.2 MAMP PRE-TRAINING (STAGE 1)

MAMP Mao et al. (2023a) uses a transformer encoder-decoder architecture to reconstruct motions from the 3D skeleton sequence. It receives the input skeleton sequence $\mathbf{S} \in \mathbb{R}^{T_s \times V \times C_s}$, where $T_s$, $V$, and $C_s$ are the temporal length, number of joints, and coordinate channels, respectively. Next, the sequence is divided into non-overlapping segments $\mathbf{S}' \in \mathbb{R}^{T_e \times V \times l \cdot C_s}$, where $T_e = T_s/l$ and $l$ is the segment length. This division results in having $T_e \times V$ tokens and reduces the temporal resolution by a factor of $l$. Subsequently, the input joints are linearly projected into joint embedding $\mathbf{E} \in \mathbb{R}^{T_e \times V \times C_e}$ where $C_e$ is the dimension of embedding features.

As for the pretraining objective and the masking strategy, MAMP leverages the motion information. Given an original sequence $\mathbf{S}$, the motion $\mathbf{M} \in \mathbb{R}^{T_s \times V \times C_s}$ is derived by employing temporal difference on joint coordinates:

$$\mathbf{M}_{i,:,:} = \mathbf{S}_{i,:,:} - \mathbf{S}_{i-m,:,:}, \quad i \in m, m+1, ..., T_s - 1 \tag{1}$$

where the step size of the motion is controlled by the hyperparameter $m$. Specifically, MAMP uses a stride of $m = l$ to capture motion among different segments of the sequence.

For masking the input sequence based on the motion, the obtained motion $\mathbf{M}$ should have the same dimension as the segmented sequence $\mathbf{S}'$. Hence, the motion $\mathbf{M}$ is padded by replicating the sequence and further reshaped into $\mathbf{M}' \in \mathbb{R}^{T_e \times V \times l \times C_s}$. Subsequently, to signify the importance of motion in each spatio-temporal segment, the motion intensity $\boldsymbol{I}$ is calculated as follows:

$$\boldsymbol{I} = \sum_{i=0}^{l} \sum_{j=0}^{C_i} |\mathbf{M}'_{:,:,i,j}| \in \mathbb{R}^{T_e \times V}, \tag{2}$$

$$\boldsymbol{P} = \texttt{Softmax}(\boldsymbol{I}/\tau_1),$$

where $\boldsymbol{P}$ indicates the probability of masking each embedding feature, and $\tau_1$ is a temperature hyperparameter. Finally, to increase the diversity in mask selection, the Gumbel-Max trick is used:

$$\boldsymbol{G} = -\log(-\log \epsilon), \, \epsilon \in U[0,1]^{T_e \times V}, \tag{3}$$

$$idx^{\text{mask}} = \texttt{Index-of-Top-K}(\log \boldsymbol{P} + \boldsymbol{G}),$$

where $U[0,1]$ represents a uniform distribution ranging from 0 to 1, and $idx^{\text{mask}}$ denotes the masked indices.

On the joint embedding $\mathbf{E}$, spatio-temporal positional embedding is added and unmasked tokens are passed to the encoder. Following the computation of the encoder's latent representations, learnable mask tokens are inserted to them according to the mask indices $idx^{\text{mask}}$. The decoder then predicts the motion $M^{pred}$ and the reconstruction loss is computed by applying mean squared error (MSE) between the predicted motion $\mathbf{M}^{pred}$ and the reconstruction target $\mathbf{M}^{target}$, as follows:

$$\mathcal{L} = \frac{1}{|idx^{\text{mask}}|} \sum_{(i,j) \in idx^{\text{mask}}} \|(\mathbf{M}^{\text{pred}}_{i,j,:} - \mathbf{M}^{\text{target}}_{i,j,:})\|_2^2. \tag{4}$$

## 3.3 CONTRASTIVE TUNING (STAGE 2)

In the second stage, we replace the decoder with projection and prediction modules. The projection module aligns the encoder representation with a space targeted for contrastive loss. The prediction module takes one positive sample from a pair and generates a representation vector resembling the other sample in the positive pair to minimize the contrastive loss. More specifically, The encoder $f_\theta$ receives segmented sequence tokens $\mathbf{S}'$ and outputs representation tokens $\mathbf{Y}_\theta = f_\theta(\mathbf{S}')$. After

applying average pooling of the output tokens, the projector $g_\theta$ aligns the result to the final representation vector $z_\theta = g_\theta(\mathbf{Y}_\theta)$. Following the NNCLR approach, vector $z_\theta$ is inserted into the queue $Q$ and is compared to sequence representations from previous iterations. From these representations, the top nearest neighbor is sampled as a positive sample in contrastive loss:

$$\text{NN}(z, Q) = \arg\min_{q \in Q} ||z - q||_2 \tag{5}$$

Concurrently, the feature vector $z_\theta$ is given to predictor module to output the feature $z_\theta^+$. Next, given positive pairs $(\text{NN}(z, Q), z^+)$, we have:

$$\mathcal{L}_i^{\text{NNCLR}} = -\log \frac{\exp\left(\text{NN}(z_i, Q) \cdot z_i^+ / \tau_2\right)}{\sum_{k=1}^n \exp\left(\text{NN}(z_i, Q) \cdot z_k^+ / \tau_2\right)} \tag{6}$$

where $\tau_2$ is a fixed temperature hyperparameter, $i$ is the sample index in batch of data, and $n$ is the batch size. Notably, in contrast to other contrastive learning approaches, our method operates more effectively with a single, unaltered view of the sequence, without relying on two different augmented views. Additionally, we show (later, in Fig. 3) that after training these two modules, the predictor output representation forms better cluster separation compared to the encoder trained with MAMP framework in previous stage.

In addition to projector and predictor modules, we partially tune the encoder parameters to produce well-separated clusters. Specifically, we use layer-wise learning rate decay Clark et al. (2020) to tune the second-half of the encoder parameters. This is formulated as:

$$LR_i = BaseLR * \alpha^{(N-i)} \tag{7}$$

where $LR_i$ denotes learning rate of the $i^{\text{th}}$ layer, $\alpha$ is the learning rate decay, and $N$ is the total number of layers.

## 4 EXPERIMENTS

### 4.1 DATASETS

**NTU-RGB+D 60 Shahroudy et al. (2016)** is a large-scale dataset containing 56,880 3D skeleton sequences of 40 subjects performing 60 actions. In this study, we use the recommended cross-subject (X-sub) and cross-view (X-view) evaluation protocols. In the cross-subject scenario, half of the subjects are selected for the training set, and the remaining subjects are used for testing. For the cross-view evaluation, sequences captured by cameras 2 and 3 are employed for training, while camera 1 sequences are used for testing.

**NTU-RGB+D 120 Liu et al. (2019)** is the extended version of NTU-60, in which 106 subjects perform 120 actions in 114,480 skeleton sequences. The authors also substitute the cross-view evaluation protocol with cross-setup (X-set), where sequences are divided into 32 setups based on camera distance and background. Samples from half of these setups are selected for training and the rest for testing.

**PKU-MMD Liu et al. (2017)** contains around 20,000 skeleton sequences of 52 actions. We follow the cross-subject protocol, where the training and testing sets are split based on subject ID. The dataset contains two phases: PKU-I and PKU-II. The latter is more challenging because of more noise introduced by larger view variations, with 5,332 sequences for training and 1,613 for testing.

### 4.2 EXPERIMENTAL SETUP

**Data Preprocessing:** From an initial skeleton sequence, a consecutive segment is randomly trimmed with a proportion $p$, where $p$ is sampled from the range [0.5, 1] during training and, similar to Mao et al. (2023a), remains fixed at 0.9 during testing. Subsequently, the segment is resized to a consistent length $T_s$ using bilinear interpolation. By default, $T_s$ is set to 120.

Table 1: Performance comparison on NTU-60, NTU-120, and PKU-MMD in the linear evaluation protocol. *Single-stream*: Joint. *Three-stream*: Joint+Bone+Motion. The best and second-best accuracies are in bold and underlined, respectively. * indicates that result is reproduced using our GPUs.

| Method | Input | NTU-60 | | NTU-120 | | PKU-II |
|---|---|---|---|---|---|---|
| | | XSub(%) | XView(%) | XSub(%) | XSet(%) | XSub(%) |
| *Other pretext tasks:* | | | | | | |
| LongTGAN Zheng et al. (2018) | Single-stream | 39.1 | 48.1 | - | - | 26.0 |
| *Contrastive Learning:* | | | | | | |
| ISC Thoker et al. (2021) | Single-stream | 76.3 | 85.2 | 67.1 | 67.9 | 36.0 |
| CrosSCLR Li et al. (2021) | Three-stream | 77.8 | 83.4 | 67.9 | 66.7 | 21.2 |
| AimCLR Guo et al. (2022) | Three-stream | 78.9 | 83.8 | 68.2 | 68.8 | 39.5 |
| CPM Zhang et al. (2022) | Single-stream | 78.7 | 84.9 | 68.7 | 69.6 | - |
| PSTL Zhou et al. (2023) | Three-stream | 79.1 | 83.8 | 69.2 | 70.3 | 52.3 |
| CMD Mao et al. (2022) | Single-stream | 79.4 | 86.9 | 70.3 | 71.5 | 43.0 |
| HaLP Shah et al. (2023) | Single-stream | 79.7 | 86.8 | 71.1 | 72.2 | 43.5 |
| HiCLR Zhang et al. (2023a) | Three-stream | 80.4 | 85.5 | 70.0 | 70.4 | - |
| HiCo-Transformer Dong et al. (2023) | Single-stream | 81.1 | 88.6 | 72.8 | 74.1 | 49.4 |
| SkeAttnCLR Hua et al. (2023) | Three-stream | 82.0 | 86.5 | 77.1 | 80.0 | **55.5** |
| I$^2$MD Mao et al. (2023b) | Three-stream | 83.4 | 88.0 | 73.1 | 74.1 | 49.0 |
| ActCLR Lin et al. (2023) | Three-stream | 84.3 | 88.8 | 74.3 | 75.7 | - |
| UmURL Sun et al. (2023) | Single-stream | 82.3 | 89.8 | 73.5 | 74.3 | 52.1 |
| *Masked Prediction:* | | | | | | |
| SkeletonMAE Wu et al. (2023) | Single-stream | 74.8 | 77.7 | 72.5 | 73.5 | 36.1 |
| MAMP Mao et al. (2023a) | Single-stream | 84.9 | 89.1 | 78.6 | 79.1 | 52.0* |
| *Masked Prediction + Contrastive Learning:* | | | | | | |
| PCM$^3$ Zhang et al. (2023b) | Single-stream | 83.9 | 90.4 | 76.5 | 77.5 | 51.5 |
| **STARS-3stage (Ours)** | Single-stream | 86.3 | 90.7 | 79.3 | 80.6 | 52.2 |
| **STARS (Ours)** | Single-stream | **87.1** | **90.9** | **79.9** | **80.8** | 52.7 |

Table 2: Performance comparison on NTU-60, NTU-120, and PKU-MMD in the KNN evaluation protocol (K=1).

| Method | NTU 60 | | NTU 120 | |
|---|---|---|---|---|
| | XSub(%) | XView(%) | XSub(%) | XSet(%) |
| P&C Su et al. (2020) | 50.7 | 75.3 | 42.7 | 41.7 |
| ISC Thoker et al. (2021) | 62.5 | 82.6 | 50.6 | 52.3 |
| MAMP Mao et al. (2023a) | 63.1 | 80.3 | 51.8 | 56.1 |
| CrosSCLR-B Li et al. (2021) | 66.1 | 81.3 | 52.5 | 54.9 |
| CMD Mao et al. (2022) | 70.6 | 85.4 | 58.3 | 60.9 |
| I$^2$MD Mao et al. (2023b) | 75.9 | 83.8 | 62.0 | 64.7 |
| **STARS-3Stage (Ours)** | 76.9 | 88.0 | 65.7 | **68.0** |
| **STARS (Ours)** | **79.9** | **88.6** | **67.6** | 67.7 |

**Network Architecture:** We adpoted the same network architecture as MAMP Mao et al. (2023a). It uses a vanilla vision transformer (ViT) Dosovitskiy et al. (2020) as the backbone with $L_e = 8$ transformer blocks and temporal patch size of 4. In each block, the embedding dimension is 256, number of multi-head attentions is 8, and hidden dimension of the feed-forward network is 1024. It also incorporates two spatial and temporal positional embeddings into the embedded inputs. The decoder used in first stage is similar to the transformer encoder except that it has $L_d = 5$ layers. In the contrastive tuning modules used in the second stage, the projector module is solely a Batch Normalization Ioffe & Szegedy (2015), given the relatively small size of the 256-dimensional embedding space. The predictor module consists of a feed-forward network with a single hidden layer sized at 4096.

**Pre-training:** The first stage follows the same setting as MAMP Mao et al. (2023a). For the second stage, we use the AdamW optimizer with weight decay 0.01, betas (0.9, 0.95), and learning rate 0.001. In the second stage, we train the projection and prediction modules in addition to finetuning the encoder for 20 epochs, and the best representation is chosen based on K-NN (K=10) on validation data. We employ layer-wise learning rate decay with a decay rate of 0.20. All the pretraining experiments are conducted using PyTorch on four NVIDIA A40 GPUs with a batch size of 32 per GPU.

Table 3: Performance comparison on NTU-60, NTU-120, and PKU-MMD in terms of the fine-tuning protocol. The best and second-best accuracies are in bold and underlined, respectively. * TF stands for Transformer.

| Method | Input | Backbone | NTU 60 | | NTU 120 | |
| --- | --- | --- | --- | --- | --- | --- |
| | | | XSub(%) | XView(%) | XSub(%) | XSet(%) |
| *Other pretext tasks:* | | | | | | |
| Colorization Yang et al. (2021) | Three-stream | DGCNN | 88.0 | 94.9 | - | - |
| Hi-TRS Chen et al. (2022) | Three-stream | Transformer | 90.0 | 95.7 | 85.3 | 87.4 |
| *Contrastive Learning:* | | | | | | |
| CPM Zhang et al. (2022) | Single-stream | ST-GCN | 84.8 | 91.1 | 78.4 | 78.9 |
| CrosSCLR Li et al. (2021) | Three-stream | ST-GCN | 86.2 | 92.5 | 80.5 | 80.4 |
| I$^2$MD Mao et al. (2023b) | Single-stream | GCN-TF* | 86.5 | 93.6 | 79.1 | 80.3 |
| AimCLR Guo et al. (2022) | Three-stream | ST-GCN | 86.9 | 92.8 | 80.1 | 80.9 |
| ActCLR Lin et al. (2023) | Three-stream | ST-GCN | 88.2 | 93.9 | 82.1 | 84.6 |
| HYSP Franco et al. (2023) | Three-stream | ST-GCN | 89.1 | 95.2 | 84.5 | 86.3 |
| *Masked Prediction:* | | | | | | |
| SkeletonMAE Wu et al. (2023) | Single-stream | STTFormer | 86.6 | 92.9 | 76.8 | 79.1 |
| SkeletonMAE Yan et al. (2023) | Single-stream | STRL | 92.8 | 96.5 | 84.8 | 85.7 |
| MAMP Mao et al. (2023a) | Single-stream | Transformer | 93.1 | **97.5** | **90.0** | 91.3 |
| *Masked Prediction + Contrastive Learning:* | | | | | | |
| W/o pre-training | Single-stream | Transformer | 83.1 | 92.6 | 76.8 | 79.7 |
| **STARS-3stage (Ours)** | Single-stream | Transformer | **93.2** | **97.5** | 89.8 | 91.3 |
| **STARS (Ours)** | Single-stream | Transformer | 93.0 | **97.5** | 89.9 | **91.4** |

## 4.3 EVALUATION AND COMPARISON

In all evaluation protocols, we report on STARS, the method proposed in section 3, as well as STARS-3stage. STARS-3stage involves a three-stage pretraining process. The second stage is divided into two parts: the Head Initialization stage, where only the projector and predictors are trained, and the contrastive tuning stage, where the encoder is fine-tuned along with the head modules. More details can be found in the supplementary materials.

**Linear Evaluation Protocol:** In this protocol, the weights of the pretrained backbone are frozen and a linear classifier is trained with supervision to evaluate the linear-separability of the learned features. We train the linear classifier for 100 epochs with a batch size of 256 and a learning rate of 0.1, which is decreased to 0 by a cosine decay schedule. We evaluate the performance on the NTU-60, NTU-120, and PKU-II datasets. As shown in Tab. 1, our proposed STARS outperforms other methods on both NTU benchmarks. On the PKU-II dataset, STARS achieves second-best result, and SkeAttnCLR Hua et al. (2023) outperforms it using a three-stream input method.

**KNN Evaluation Protocol:** An alternative way to evaluate the pretrained encoder is by directly applying a K-Nearest Neighbor (KNN) classifier to their output features. Following other works Su et al. (2020); Mao et al. (2022; 2023b), each test sequence is compared to all training sequences using cosine similarity and the test prediction is based on the label of the most similar neighbor (i.e. KNN with k=1). Tab. 2 compares different methods using KNN evaluation protocol. Notably, we find that MAMP cannot achieve competitive results compared to contrastive learning models, despite showing superior results on linear evaluation. We believe that this is because of the pretraining objective of contrastive learning models, which, by pushing different samples into different areas of the representation space, results in better-separated clusters. Our STARS approach leverages contrastive tuning to enhance the feature representation of MAMP, outperforming all other methods. This demonstrates the superiority of contrastive tuning over contrastive learning approaches.

**Fine-tuned Evaluation Protocol:** We follow MAMP and by adding MLP head on the pretrained backbone, the whole network is fine-tuned for 100 epochs with batch size of 48. The learning rate starts at 0 and is gradually raised to 3e-4 during the initial 5 warm-up epochs, after which it is reduced to 1e-5 using a cosine decay schedule. As shown in Tab. 3, both MAMP and STARS notably enhance the performance of their transformer encoder without pretraining. However, these results indicate that contrastive tuning following MAMP pretraining does not impact the fine-tune evaluation, and MAMP and STARS achieve nearly identical results, both outperforming other approaches.

Table 4: Performance comparison in the transfer learning protocol, where the source datasets are NTU-60 and NTU-120, and the target dataset is PKU-II.

| Method | To PKU-II | |
|---|---|---|
| | NTU 60 | NTU 120 |
| LongTGAN Zheng et al. (2018) | 44.8 | - |
| MS2L Lin et al. (2020) | 45.8 | - |
| ISC Thoker et al. (2021) | 51.1 | 52.3 |
| CMD Mao et al. (2022) | 56.0 | 57.0 |
| HaLP+CMD Shah et al. (2023) | 56.6 | 57.3 |
| SkeletonMAE Wu et al. (2023) | 58.4 | 61.0 |
| MAMP Mao et al. (2023a) | 70.6 | **73.2** |
| **STARS-3stage (Ours)** | 71.8 | 72.7 |
| **STARS (Ours)** | **71.9** | 72.2 |

Table 5: Performance comparison in the few-shot settings, where the model is pretrained on NTU-60 XSub and tested on 60 new samples of NTU-120 XSub.

| Method | 1-shot | 2-shot | 5-shot |
|---|---|---|---|
| MAMP Mao et al. (2023a) | 47.6 | 44.4 | 48.4 |
| AimCLR Guo et al. (2022) | 48.9 | 45.9 | 51.1 |
| HiCLR Zhang et al. (2023a) | 51.7 | 49.6 | 53.8 |
| ISC Thoker et al. (2021) | 55.4 | 53.3 | 57.1 |
| HiCo-Transformer Dong et al. (2023) | 60.0 | 58.2 | 60.9 |
| CMD Mao et al. (2022) | 61.2 | 58.2 | 61.3 |
| **STARS-3stage (Ours)** | 59.3 | 57.8 | 61.5 |
| **STARS (Ours)** | **63.5** | **62.2** | **65.7** |

**Transfer Learning Protocol:** In this protocol, the transferability of the learned representation is evaluated. Specifically, the encoder undergoes pretraining on a source dataset using a self-supervised approach, followed by fine-tuning on a target dataset through a supervised method. In this study, NTU-60 and NTU-120 are selected as the source datasets, with PKU-II chosen as the target dataset. Tab. 4 shows that when fine-tuned on a new dataset, masked prediction techniques like Skeleton-MAE and MAMP demonstrate superior transferability compared to contrastive learning methods. Moreover, STARS enhances performance when pre-trained on NTU-60, but its effectiveness diminishes when pre-trained on NTU-120.

**Few-shot Evaluation Protocol:** This protocol evaluates the scenario where only a small number of samples are labeled in the target dataset. This is crucial in practical applications like education, sports, and healthcare, where actions may not be clearly defined in publicly available datasets. In this protocol, we pretrain the model on NTU-60 (XSub) and evaluate it on the evaluation set of 60 novel actions on NTU-120 (XSub) using $n$ labeled sequences for each class in $n$-shot setting. For the evaluation, we follow MotionBERT Zhu et al. (2023) and calculate the cosine distance between the test sequences and the exemplars, and use $n$-nearest neighbors to determine the action. Tab. 5 compares different methods in the few-shot settings. Notably, MAMP demonstrates poor generalization performance, in contrast to its robust performance in transfer learning and evaluations within the dataset. By applying contrastive tuning, STARS surpasses contrastive learning approaches in all settings, demonstrating its strength in various evaluations.

**Qualitative Comparison:** Fig. 3 compares the t-SNE visualization of our proposed STARS method with AimCLR Guo et al. (2022), CMD Mao et al. (2022), HiCo-Transformer Dong et al. (2023), and MAMP Mao et al. (2023a). CMD adds cross-modal mutual distillation loss to contrastive learning and by ensuring that various input modalities (joint, bone, and motion) exhibit the same neighborhood, it scatters actions across different areas of space and mitigates the impact of applying contrastive learning loss. On the other hand, AimCLR and HiCo-Transformer create distinct clusters through the use of extreme augmentations and by applying contrastive loss at different hierarchical levels, respectively. When compared to MAMP, these two contrastive learning methods exhibit a higher inter-cluster distance than MAMP. Interestingly, actions involving interactions between two individuals, such as kicking, giving objects, and shaking hands, create a distinct higher-level cluster compared to actions involving a single person across various methods. Specifically, MAMP shows the highest distance between these two cluster groups, whereas within each group, the action clusters are closely situated. By employing contrastive tuning, STARS effectively minimizes intra-cluster distance (as seen in examples like sneeze/cough) while maximizing inter-cluster distance. This leads to the formation of clearly separated clusters, each representative of different actions.

## 4.4 ABLATION STUDY

**Tuning Strategy Design:** Tab. 6 compares the NNCLR strategy used in our STARS framework with DINO and MoCo. DINO Caron et al. (2021) employs a student-teacher framework. It updates the student's weights by relying on the teacher's output, which is constructed using a momentum encoder, as the target. Unlike contrastive learning methods, DINO does not need negative samples

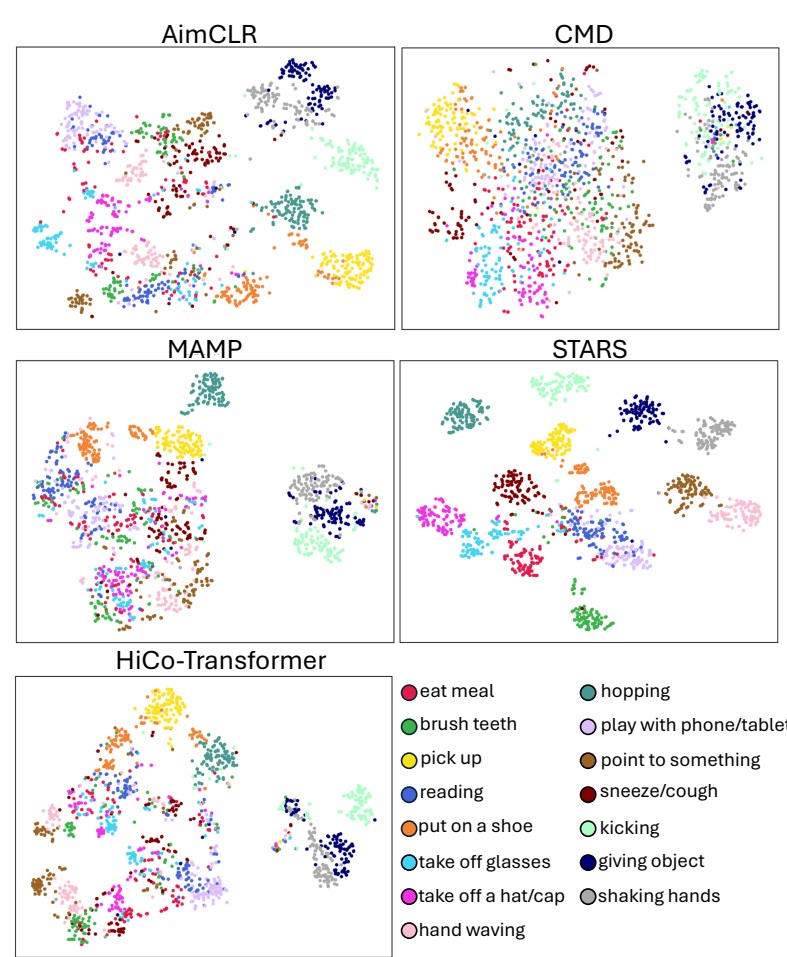

Figure 3: The t-SNE visualization of embedding features. We sample 15 action classes from the NTU-60 dataset and visualize the features extracted by our proposed STARS framework and compare it with Aim-CLR Guo et al. (2022), CMD Mao et al. (2022), HiCo-Transformer Dong et al. (2023), and MAMP Mao et al. (2023a).

for contrast and employs centring and sharpening techniques to prevent collapse. MoCo He et al. (2020) is predominantly used by other contrastive learning approaches in action recognition Li et al. (2021); Guo et al. (2022); Lin et al. (2023). It uses a memory bank to increase the negative samples in contrastive loss and a key encoder, which is updated via exponential moving average to maintain consistency. As shown in the Tab. 6, NNCLR significantly enhances KNN accuracy by forming better clusters for different actions, while not using any data augmentations. For the remaining two strategies, we also examined the impact of including augmentation through spatial flipping and rotation. Generally, adding augmentations helps the methods achieve better performance; especially for MoCo, which relies on augmentations to construct the positive samples. Note that it is expected for the other two methods to further improve by incorporating more augmentations, which is not the focus of this study. Additional details about the hyperparameters in this ablation study are provided in the supplementary material.

**Effect of Augmentation:** Tab. 7 shows that applying augmentation results in a minor improvement in the KNN evaluation protocol. However, we chose not to use augmentation as our main method since the type of augmentation works heuristically and can result in different behavior in new scenarios, sometimes even degrading performance in cases such as shearing or axis masking. Additionally, we tested data augmentation on different evaluation protocols, such as linear evaluation, and did not observe any performance improvement.

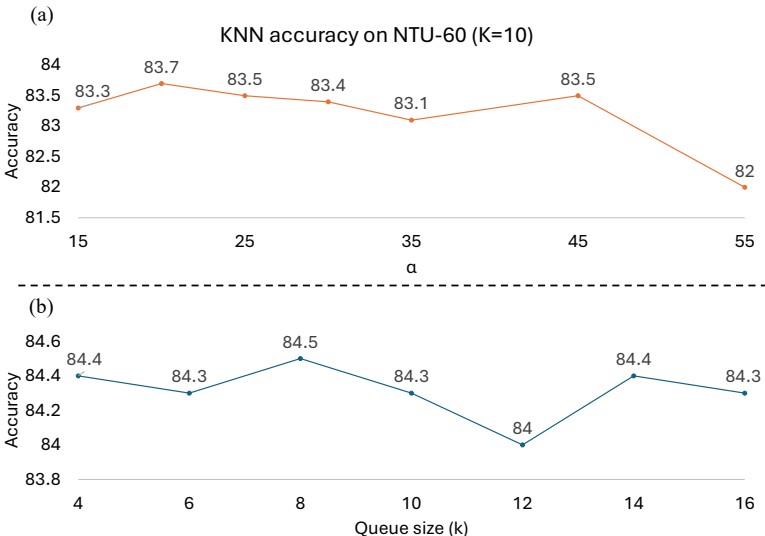

Figure 4: Ablation study on (a) layer-wise learning decay (b) Queue size. The performance is evaluated on the NTU-60 XSub dataset under the KNN evaluation protocol (K=10).

Table 6: Ablation study on the tuning strategy. The performance is evaluated on the NTU-60 XSub and NTU-60 XView datasets under the KNN evaluation protocol (K=10).

| Tuning Strategy | NTU-60 | |
| | XSub | XView |
| --- | --- | --- |
| DINO | 77.6 | 86.3 |
| DINO$_{aug}$ | 77.4 | 86.7 |
| MoCo | 72.2 | 86.7 |
| MoCo$_{aug}$ | 73.9 | 88.0 |
| **NNCLR** | **81.9** | **89.6** |

Table 7: Ablation study on the effect of augmentation. The performance is evaluated on the NTU-60 XSub and NTU-60 XView datasets under the KNN evaluation protocol (K=10).

| Augmentation | NTU-60 | |
| | XSub | XView |
| --- | --- | --- |
| Spatial Flip | **85.0** | **90.6** |
| Rotation | 84.8 | 90.1 |
| Axis Mask | 81.2 | 89.0 |
| Shear | 83.2 | 90.4 |
| Spatial Flip + Rotation | 84.6 | 90.4 |
| No Augmentation | 84.5 | 89.6 |

**Layer-wise Learning Rate Decay:** As shown in Fig. 4 (a), we observe a decrease in accuracy with higher learning decay. Our hypothesis is that increasing the decay causes the encoder to forget the robust representations learned in the initial stage, leading to performance degradation comparable to contrastive learning methods.

**Queue size:** Fig. 4 (b) explores how different queue sizes affect model accuracy during contrastive tuning, evaluated using the KNN protocol. The results indicate that the queue size has little impact on performance during pretraining. Based on these findings, we chose a queue size of 8k for all our evaluations.

## 5 CONCLUSION

In this work, we proposed a sequential contrastive tuning method. We find that masked prediction methods, despite showing promising results in various within-dataset evaluations, cannot outperform contrastive learning based methods in few-shot settings. By using our STARS framework, we show that we can further enhance the masked prediction baseline while achieving competitive results in few-shot settings, outperforming other models in 5-shot setting. However, when the dataset size is limited for pretraining and for evaluations when encoder is fine-tuned with supervision, STARS cannot add significant value to its baseline.

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

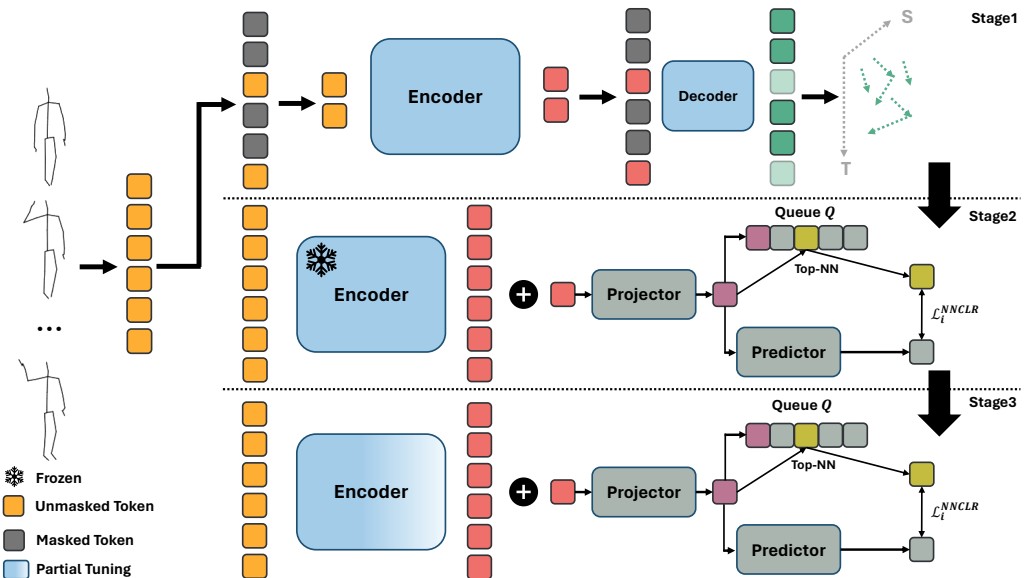

Figure 5: The overall pipeline of our proposed STARS-3stage framework. The first stage uses MAMP Mao et al. (2023a) to reconstruct the motion of masked tokens. The second stage keeps the encoder parameters frozen and trains parameters of the projector and predictor using a contrastive learning approach. After these parameters have converged to well-separated clusters, the third stage involves partial-tuning of the encoder parameters.

## A    3-STAGE DESIGN

An alternative pretraining method is to follow MAE-CT Lehner et al. (2023) and tune the encoder in 3 stages. Figure 5 shows the 3-stage design. Initially, when we initialize the projector and predictor modules, we freeze the encoder weights. In the second stage, we exclusively train the projector and predictor modules until they can effectively differentiate between different sequences using the NNCLR approach. Finally, in the third stage, we fine-tune the encoder weights using layer-wise learning rate decay. One motivation for this staged approach is the idea that the NNCLR head's random weights could interfere with the representation quality by mapping the features into a random space, disrupting the learned structure. However, our findings challenge this assumption. The t-SNE visualization in Fig. 6 demonstrates that even with random NNCLR head weights, the cluster structure in the encoder's output space remains intact in the new representation space. Furthermore, we observed with STARS-3stage that replicating this three-stage process not only increases training time but also leads to a drop in final accuracy. As a result, we use a two-stage design in our proposed STARS method.

## B    SEMI-SUPERVISED EVALUATION RESULTS

In semi-supervised evaluation protocol, we follow previous works Li et al. (2021); Mao et al. (2022; 2023a) and fine-tune the pretrained encoder in addition to a post-attached classifier while given a small fraction of the training dataset. Specifically, the performance on the NTU-60 is reported while using 1% and 10% of the training set. Since the training portions are selected randomly, we report the result averaged over 5 different runs as the final result. As shown in Tab. 8, STARS is more effective in all scenarios. Specifically, while using 1% of the training data, leading to an increase in accuracy for the MAMP baseline by 3.1% and 4.2% in cross-subject and cross-view evaluations, respectively.

## C    ABLATION HYPER-PARAMETERS

Tab. 9 shows the hyperparameters used in DINOTuning strategy. For simplicity and because of limitation in resources, we used only two global views and did not use any local views in DINO. As shown in Tab. 10, we can see that incorporating local views led to a small improvement in

MAMP before projection layer          MAMP after projection layer

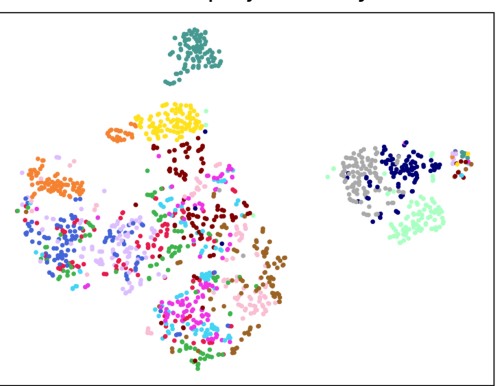

Figure 6: Comparison between MAMP's output vectors before and after using a projection layer with random weights.

Table 8: Performance comparison on the NTU-60 dataset under the semi-supervised evaluation protocol, with the final performance reported as the average of five runs.

| Method | NTU-60 | | | |
| | XSub | | XView | |
| | (1%) | (10%) | (1%) | (10%) |
|---|---|---|---|---|
| *Other pretext tasks:* | | | | |
| LongT GAN Zheng et al. (2018) | 35.2 | 62.0 | - | - |
| ASSL Si et al. (2020) | - | 64.3 | - | 69.8 |
| *Contrastive Learning:* | | | | |
| MS$^2$L Lin et al. (2020) | 33.1 | 65.1 | - | - |
| ISC Thoker et al. (2021) | 35.7 | 65.9 | 38.1 | 72.5 |
| 3s-CrosSCLR Li et al. (2021) | 51.1 | 74.4 | 50.0 | 77.8 |
| 3s-Colorization Yang et al. (2021) | 48.3 | 71.7 | 52.5 | 78.9 |
| CMD Mao et al. (2022) | 50.6 | 75.4 | 53.0 | 80.2 |
| 3s-Hi-TRS Chen et al. (2022) | 49.3 | 77.7 | 51.5 | 81.1 |
| 3s-AimCLR Guo et al. (2022) | 54.8 | 78.2 | 54.3 | 81.6 |
| 3s-CMD Mao et al. (2022) | 55.6 | 79.0 | 55.5 | 82.4 |
| CPM Zhang et al. (2022) | 56.7 | 73.0 | 57.5 | 77.1 |
| *Masked Prediction:* | | | | |
| SkeletonMAE Wu et al. (2023) | 54.4 | 80.6 | 54.6 | 83.5 |
| MAMP Mao et al. (2023a) | 66.0 | 88.0 | 68.7 | 91.5 |
| *Masked Prediction + Contrastive Learning:* | | | | |
| PCM$^3$ Zhang et al. (2023b) | 53.1 | 82.8 | 53.8 | 77.1 |
| **STARS-3stage (Ours)** | 68.6 | **88.2** | 72.5 | **91.8** |
| **STARS (Ours)** | **69.1** | 88.0 | **72.9** | **91.8** |

performance. However, it came at the cost of significantly more resources. To be specific, we introduced two local views that randomly trimmed a section of the sequence between 40% and 80% and fed it only to the student network. With these additional views, we had to reduce the batch size to 16 and double the training time. Consequently, in our other experiments, we stuck to using only global views. We also used Sinkhorn-Knopp centering Caron et al. (2020) a KoLeo regularizer Sablayrolles et al. (2018) to help the convergence. Tab 15 shows the hyperparameters used in MoCoTuning. Similar to previous approaches Guo et al. (2022); Li et al. (2021), we used 32K as the queue size, 0.999 for the momentum and 0.07 for the temperature.

Table 9: DINOTuning hyperparameters for ablation study in tuning strategy design.

| Hyperparameter | Value |
| --- | --- |
| Learning rate | 0.001 |
| Batch size | 32 |
| Augmentations | Mirroring & Rotation |
| Centering | Sinkhorn Knopp |
| KoLeo weight | 0.1 |
| # Global views | 2 |
| # Local views | 0 |
| Student temperature | 0.1 |
| Teacher temperature | 0.04 |
| Teacher momentum | 0.996 |

**Algorithm 1** PyTorch-style pseudo-code of contrastive tuning in the second stage.

```
1   # f: MAMP Encoder. Only second-half is tuned.
2   # g: Projector. Batch normalization module
3   # h: Predictor. 2 layer MLP, hidden size 4096, output 256
4   # Q: Queue with length of 32,768
5
6   for x in loader:
7       y = f(x) # encoder forward pass
8       z = g(y) # projection forward pass
9       p = h(z) # prediction forward pass
10      z, p = normalize(z, dim=1), normalize(p, dim=1)
11      nn = top_nn(z, Q) # finding nearest-neighbor sample in Q
12      loss = L(nn, p)
13      loss.backward()
14      optimizer.step()
15      update_queue(Q, z)
16
17
18  def top_nn(z, Q):
19      similarities = z @ Q.T
20      idx = similarities.max(dim=1)
21      return Q[idx]
22
23  def L(nn, p, temperature=0.07):
24      logits = nn @ p.T
25      logits /= temperature # sharpening
26      labels = torch.arange(p.shape[0])
27      loss = cross_entropy(logits, labels)
28      return loss
```

Table 10: Effect of including local views on DINOTuning.

| Method | 10-NN | 20-NN | 40-NN |
| --- | --- | --- | --- |
| w/o local views | 77.4 | 77.1 | 77.0 |
| w/ local views | 77.8 | 78.0 | 77.6 |

## C.1 ALGORITHM PSUEDO CODE

Algo. 1 demonstrates the process in PyTorch-style pseudo code.

## D ADDITIONAL ABLATION STUDIES

**Combining MAMP and NNCLR.:** One idea to tune the encoder in second stage is to combine NNCLR with MAMP and use both. Tab 11 compares this tuning stage with the proposed STARS. By including MAMP in the second stage of tuning, although it improves the final representation of encoder compared to the baseline (MAMP), it cannot perform as effective as STARS.

**Training NNCLR from scratch:** Table 12 presents the K-NN evaluation results for training the transformer from scratch using the NNCLR method for 300 epochs. Without augmentation, the model struggles to select positive samples in contrastive learning that truly represent the same ac-

Table 11: Ablation study on second stage of tuning. The performance is evaluated on the NTU-60 XSub and NTU-60 XView datasets under the KNN evaluation protocol (K=1).

| Method | NTU-60 | |
| --- | --- | --- |
| | XSub | XView |
| MAMP (Baseline) | 63.1 | 80.3 |
| STARS | **79.9** | **88.6** |
| MAMP + NNCLR | 74.6 | 86.5 |

Table 12: Training the transformer encoder using NNCLR method from scratch. The performance is evaluated on the NTU-60 XSub and NTU-60 XView datasets under the KNN evaluation protocol

| K | NTU-60 | |
| --- | --- | --- |
| | XSub | XView |
| 1 | 37.6 | 30.5 |
| 2 | 35.8 | 28.7 |
| 5 | 39.9 | 32.3 |
| 10 | 41.2 | 33.6 |

Table 13: MoCo hyperparameters for ablation study in tuning strategy design.

| Hyperparameter | Value |
| --- | --- |
| Learning rate | 0.001 |
| Batch size | 32 |
| Augmentations | Mirroring & Rotation |
| Queue size | 32,768 |
| Momentum | 0.999 |
| Temperature | 0.07 |

Table 14: Comparison on memory usage in pretraining between the methods.

| Method | Memory usage (MB) |
| --- | --- |
| MAMP | 240 |
| AimCLR | 139 |
| ActCLR | 116 |
| CMD | 1,492 |
| STARS | 998 |

tions. This happens because the encoder starts with random weights, which lack meaningful cluster separation. Consequently, the encoder fails to fully use the advantages of NNCLR in the second stage of STARS, resulting in poorer performance.

**Memory usage:** Table **??** compares memory usage across different methods using a single input (Batch size = 1) and observed notable differences. MAMP, a transformer-based approach, generally consumes more memory due to its complexity but mitigates this by processing only 10% of tokens in the encoder and reconstructing the mask using a lightweight decoder. In contrast, STARS processes all tokens and incorporates queues for contrastive learning, resulting in higher memory usage. AimCLR and ActCLR, which are GCN-based, require significantly less memory. CMD, utilizing three encoders for joint, motion, and bone streams along with a GRU-based design, demonstrates the highest memory consumption.

**Using naive MAE instead of MAMP:** Tab. 14 and Tab.16 present a comparison of K-NN and few-shot evaluations for a variant that uses naive MAE instead of MAMP in the first stage. While tuning in the second stage leads to significant improvements, the overall performance remains lower because MAE performs worse than MAMP in the initial stage.

# E   CONFUSION MATRIX

Fig. 7 illustrates the confusion matrix under KNN evaluation protocol when K=10 on NTU-60 XSub dataset. The errors depicted in the figure can be classified into two distinct categories. Firstly, there are errors stemming from a lack of contextual information. For instance, when only a skeletal sequence is provided, actions like "play with phone/tablet" might be misinterpreted as "reading" and "writing." Secondly, there are errors arising from subtle movements, such as distinguishing between "clapping" and "hand rubbing," which pose challenges for the model in differentiation. In summary, these errors manifest due to either insufficient context or the intricacy of distinguishing minute actions, highlighting the complexities inherent in the task.

Table 15: Ablation study on first K-NN evaluation by changing the first-stage of STARS to naive MAE.

| Method | NTU-60 | |
| --- | --- | --- |
| | XSub | XView |
| MAMP | 63.1 | 80.3 |
| STARS | 79.9 | 88.6 |
| MAE | 44.1 | 43.7 |
| STARS-MAE | 53.5 | 55.2 |

Table 16: Ablation study on first few-shot evaluation by changing the first-stage of STARS to naive MAE.

| Method | 1-shot | 2-shot | 5-shot |
| --- | --- | --- | --- |
| MAMP | 47.6 | 44.4 | 48.4 |
| STARS | 63.5 | 62.2 | 65.7 |
| MAE | 35.0 | 31.8 | 34.2 |
| STARS-MAE | 41.5 | 37.9 | 40.6 |

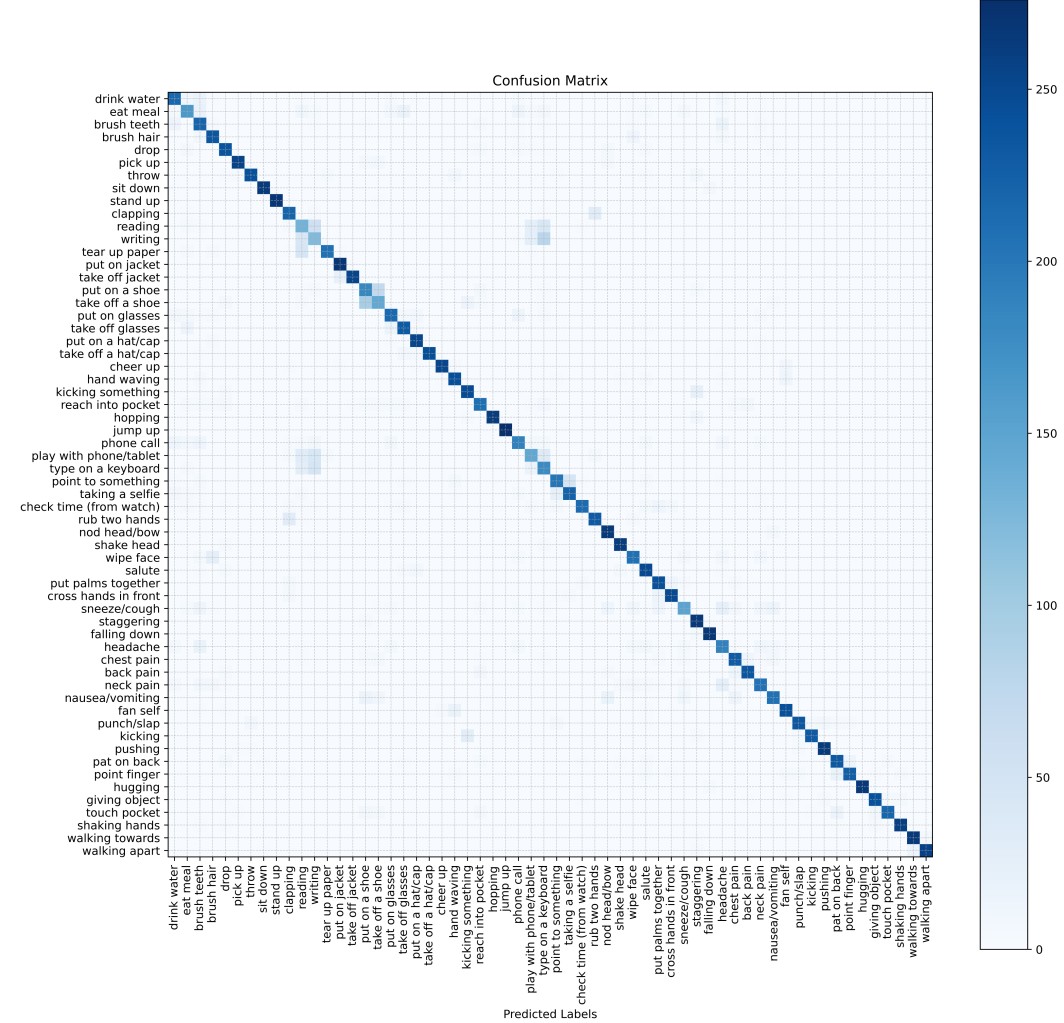

Figure 7: Confusion matrix of KNN evaluation in NTU-60 XSub dataset (k=10).