# OpenReview forum: "STARS: Self-supervised Tuning for 3D Action Recognition in Skeleton Sequences"
_ICLR.cc/2025/Conference — Submitted to ICLR 2025_

### Official Review · Reviewer_XCkD · 2024-10-24

**Soundness:** 2
**Presentation:** 2
**Contribution:** 2
**Rating:** 5
**Confidence:** 4

**Summary:**

This paper proposes an STARS framework that combines Masked Auto-Encoding (MAE) and contrastive learning to perform self-supervised tuning for 3D action recognition. It utilizes MAE as the pretext task and trains a contrastive head to partially tune the encoder, so as to learn distinct clusters for better action recognition. Experiments demonstrate the effectiveness of STARS on various benchmarks, including NTU-60, NTU-120, and PKU-MMD. This work also shows the limited generalizability of MAE approaches in few-shot settings, and verifies the higher efficacy of STARS under different protocols (linear evaluation, KNN evaluation, fine-tuned evaluation, etc.) in most cases.

**Strengths:**

1. This paper empirically compares the performance of masked prediction methods (MAE) and contrastive learning methods on action recognition, and conceptually introduces nearest-neighbor contrastive learning into MAE by partial self-supervised tuning to enhance the generalization performance. This idea is simple and effective.

2. This work conducts relatively comprehensive experiments under different cases, covering conventional protocols and new few-shot evaluation protocol, and compares with existing state-of-the-arts to show the effectiveness of method.

3. The paper writing is easy to follow with clear technical description and presentation.

**Weaknesses:**

1. This work lacks sufficient novelty. It seems that the authors stack and combine existing technologies, such as MAE and contrastive learning approaches, to build the proposed framework for 3D action recognition.

2. The comparison with other models in Fig. 1 is vague and not specific. The authors do not provide any information about training parameter size, training computational complexity (e.g., GFLOPs), etc. The training time can be influenced by many factors such as machines, GPU ability, I/O speed, etc. So I do not think the comparison in Fig. 1 can fairly show the efficiency or resource usage of the proposed method, unless providing a comprehensive description about training process of each model or the above metrics.

3. This work claims that "MAE approaches exhibit a lack of generalizability in few-shot settings", but it lacks sufficient experiments and deeper analyses to support this crucial claim. Authors seem to use only MAMP as an example of MAE approaches to compare in few-shot settings (Sec. 4.3), but other MAE methods are not included in the comparison and analyses. To provide a thorough evaluation of generalizability, it is suggested to add more empirical comparison (e.g., the naive MAE and its different representative variants), more qualitative analysis (e.g., the generalization performance of learned features on similar or different action classes), and more evaluation scenarios (e.g., other datasets rather than only NTU-60 and NTU-120).

4. It is suggested to quantify the required number of epochs instead of using "a few epochs" in the contribution part. Is the number of epochs fixed or does it need to be adjusted according to different scenarios? All these questions should be clearly clarified in the paper, for a more solid and improved presentation of this paper.

5. The paper mentioned that "any alternative MAE-based approach is also applicable", and stated that MAMP shows promising results. How is the performance of other MAE-based approaches when applied to the proposed framework? As we know, the generality of a method is not only related to a certain evaluation protocol such as the few-shot settings, but also related to its general applicability under different cases. Therefore, it is important to add compared experiments using different MAE-based approaches (there are many MAE methods in this area).

6. Some details of technical components should be added with their motivations. For example, what is the reason for "use layer-wise learning rate decay Clark et al. (2020) to tune the second-half of the encoder parameters"? The authors add this component but do not explain the necessity or importance.

7. Some results in the qualitative comparison seem problematic and not convincing. The result "52.0*" of MAMP evaluated on PKU-II (Table 1) is not the same as the original paper, do different GPUs influence the performance? Authors do not present the best result of CMD (Mao et al 2022), i.e., Three-stream as input, in Table 1. Authors need to carefully check and comprehensively and farily present all published results. The t-SNE visualization in Figure. 3 is also different from other papers, such as CMD. In the CMD paper, the visualization result is significantly better than that shown in this paper.

8. Some result points are missed in the ablation study (Figure 4). For example, the paper only shows the KNN accuracy of α=15, 25, 35, 55, but lacks the point when setting α=45. For queue size experiments, what are the results when setting k=6, 10, 12, 14? Authors should add these results to get a more holistic view of ablation study.

**Questions:**

1. Please provide a comprehensive description about training process of each model (in Fig. 1) or adopt more metrics, as detailed in the Weakness 2.

2. To provide a thorough evaluation of generalizability, it is suggested to add more empirical comparison (e.g., the naive MAE and its different representative variants), more qualitative analysis (e.g., the generalization performance of learned features on similar or different action classes), and more evaluation scenarios (e.g., other datasets rather than only NTU-60 and NTU-120). (see Weakness 3)

3.  It is suggested to quantify the required number of epochs instead of using "a few epochs" in the contribution part. Is the number of epochs fixed or does it need to be adjusted according to different scenarios? All these questions should be clearly clarified in the paper, for a more solid and improved presentation of this paper.

4.  How is the performance of other MAE-based approaches when applied to the proposed framework? (detailed in Weakness 5)

5. What is the reason for "use layer-wise learning rate decay Clark et al. (2020) to tune the second-half of the encoder parameters"? (see Weakness 6)

6. The result "52.0*" of MAMP evaluated on PKU-II (Table 1) is not the same as the original paper, do different GPUs influence the performance? (see Weakness 7)

7. The authors should provide comprehensive results in the ablation study. (detailed in Weakness 8)

---

> ### Author Response · Authors · 2024-11-22
>
> > Please provide a comprehensive description about training process of each model (in Fig. 1) or adopt more metrics, as detailed in the Weakness 2.
>
> In Fig. 1, we used the codes provided by each work to train for a single epoch and estimated the training time by multiplying it by the number of epochs provided by each work. In general:
> - MAMP although uses a Transformer backbone and it is more costly to be trained, 90% of tokens are masked so the heavy encoder only receives 10% of the data and then a lightweight decoder has to reconstruct the motion for the remaining 90%, leading to a fast training time.
> - STARS adds a bit of training time in the second stage (first stage is MAMP) by tuning the encoder by only 20 epochs. That’s the main reason why it is faster than other contrastive learning based methods since they train from scratch. More specifically, AimCLR and ActCLR train for 300 epochs, and CMD trains for 450 epochs.
>
> > To provide a thorough evaluation of generalizability, it is suggested to add more empirical comparison ...
>
> &
>
> > How is the performance of other MAE-based approaches when applied to the proposed framework?
>
> Since weights of a naive MAE model from MAMP work is not shared, we retrained the model from scratch using a naive MAE objective. We also trained STARS using MAE in the first stage. The tables below show their K-NN evaluation (K=1) and in few-shot settings.
>
> K-NN evaluation:
> | Method      | NTU-60 XSub (%) | NTU-60 XView (%) |
> |-------------|-----------------|------------------|
> | MAMP        | 63.1            | 80.3            |
> | STARS       | 79.9            | 88.6            |
> | MAE         | 44.1            | 43.7            |
> | STARS-MAE   | 53.5            | 55.2            |
>
> Few-shot:
> | Method      | 1-shot (%) | 2-shot (%) | 5-shot (%) |
> |-------------|------------|------------|------------|
> | MAMP        | 47.6       | 44.4       | 48.4       |
> | STARS       | 63.5       | 62.2       | 65.7       |
> | MAE         | 35.0       | 31.8       | 34.2       |
> | STARS-MAE   | 41.5       | 37.9       | 40.6       |
>
> We also added these two tables in the revised supplementary material.
>
> > It is suggested to quantify the required number of epochs instead of using "a few epochs" in the contribution part.
>
> We have mentioned the number of epochs on section 4.2 - pretraining. It needs a maximum of 20 epochs and the best representation is chosen based on K-NN (K=10) on validation data. We added this detail in the revised version. Thank you for mentioning it.
>
> > What is the reason for "use layer-wise learning rate decay Clark et al. (2020) to tune the second-half of the encoder parameters"?
>
> Please refer to the answer provided to the third question of reviewer iumL.
>
> > The result "52.0*" of MAMP evaluated on PKU-II (Table 1) is not the same as the original paper, do different GPUs influence the performance?
>
> Yes. The checkpoints from the PKU-II dataset is not provided by the MAMP authors. So we had to reproduce it by training using their provided code. However, probably due to differences in the GPUs used, we couldn’t reproduce the results reported in their paper.
>
> > The authors should provide comprehensive results in the ablation study.
>
> Thank you for mentioning the missed ablations. We updated the Figure 4 of the paper to include the mentioned results. For the choice of K in K-NN accuracy, typically only K=1 is used in the literature. We also provided K=10 in our ablation as we used that for tuning the hyperparameters.

---

> > ### Comment · Reviewer_XCkD · 2024-11-30
> >
> > Thank you for providing more experimental results in combining naive MAE model and the the proposed STARS, and I appreciate the answers to my questions from the authors, which have partially addressed my concerns.
> >
> > However, as mentioned in my second question, I think the generalizability evaluation of the method is limited, i.e., insufficient experimental results to demonstrate its wide applicability to other backbone, benchmarks, or tasks. Considering that the authors only provide the experimental results related to naive MAE, it is suggested to add more related experiments to show its generality.
> >
> > The explanation for some results (such as in question 6) is not convincing. As the MAMP results have been published in the original paper, it is recommended to directly compare them instead of reproducing a "lower" result, which might cause some confusion and unfairness.
> >
> > Based on the above consideration, I agree with the Reviewer qu9p that the contributions and generality of this work are limited to the related communities, and I think this work may lack sufficient novelty (as stated in the Weakness 1). Thus, I choose to maintain the current score.
> >
> > Thanks

---

> > > ### Author Response · Authors · 2024-12-01
> > >
> > > Thank you for your feedback and review. Regarding the main concerns you stated:
> > >
> > > > I think the generalizability evaluation of the method is limited, i.e., insufficient experimental results to demonstrate its wide applicability to other backbone, benchmarks, or tasks.
> > >
> > > - **Backbone:**  The STARS framework focuses on refining the representations of a transformer-based model that has been pretrained using a masked autoencoder (MAE) approach. Unlike traditional backbones like ST-GCN, which lack a global receptive field, transformers are better suited for this purpose. Therefore, it wouldn’t make sense to use backbones like ST-GCN for STARS.
> > > - **Benchmarks:** Most self-supervised learning methods in this field evaluate their performance on datasets like NTU-60, NTU-120, and PKU. While some methods skip certain evaluations, such as semi-supervised comparisons, STARS has been tested comprehensively, including these and a few-shot evaluation. This few-shot evaluation had been previously overlooked but reveals that MAE approaches underperform compared to contrastive learning methods. STARS, however, delivers a significant improvement in this scenario.
> > > - **Tasks**: STARS is specifically designed as a two-stage pretraining approach for action recognition. Unlike contrastive learning methods, which often rely on data augmentation (and risk misleading the model in action recognition), STARS avoids augmentation entirely, making it better suited for this task.
> > >
> > > > it is recommended to directly compare them instead of reproducing a "lower" result, which might cause some confusion and unfairness.
> > >
> > > The MAMP method does not provide checkpoints for the PKU-II benchmark. As a result, we had to pretrain the model from scratch, which meant we couldn’t reproduce their reported results. What shown in Table 1 is the performance gain when STARS refinement stage is applied to MAMP and we had to show the reproduced result (52.0%) to emphasize the performance gain that STARS has made.
> > >
> > > We hope the explanations provided above address any ambiguities, and we kindly request you to reconsider your review score.

---

> ### Comment · Reviewer_XCkD · 2024-12-02
> **Final Evaluation**
>
> Thanks for the authors' response. However, the answer is not convincing for the following reasons:
>
> >"Backbone: The STARS framework focuses on refining the representations of a transformer-based model that has been pretrained using a masked autoencoder (MAE) approach. Unlike traditional backbones like ST-GCN, which lack a global receptive field, transformers are better suited for this purpose. Therefore, it wouldn’t make sense to use backbones like ST-GCN for STARS."
>
> It seems that the STARS framework can only be used on transformer-based models with limited generality for other paradigms. There should be some backbones besides multi-scale GCN and its variants [1][2] that can utilize multi-scale receptive fields or global receptive fields to learn action representations. I suggest the authors to carefully check the literature and evaluate the framework on these models for a more thorough validation of method generality.
>
> >"Benchmarks: Most self-supervised learning methods in this field evaluate their performance on datasets like NTU-60, NTU-120, and PKU. While some methods skip certain evaluations, such as semi-supervised comparisons, STARS has been tested comprehensively, including these and a few-shot evaluation. This few-shot evaluation had been previously overlooked but reveals that MAE approaches underperform compared to contrastive learning methods. STARS, however, delivers a significant improvement in this scenario."
>
> I acknowledge that few-shot evaluation is a relatively new evaluation setting in this work. However, **adding new evaluation protocols does NOT mean improving the novelty of the proposed method**. I agree with Reviewer qu9p and have concerns about the **insufficient novelty** of this work. For the generalizability of this method, I still think a more comprehensive evaluation should be conducted, including more action recognition benchmark datasets such as NW-UCLA and more up-to-date MAE-based approaches (as the authors seem to claim that all MAE approaches underperform).
>
> I am not satisfied with the response of the authors to each weakness point (such as point 7), and **I hope the authors can carefully read and revise the current paper point by point**.
>
> I provide a final rating of 5, which can not be improved considering the overall contributions, novelty, and generality. I’ll leave the final decision of this matter to AC.
>
> Thanks.
>
>
> References:
>
> [1] Zhang et al. Context Aware Graph Convolution for Skeleton-Based Action Recognition. CVPR, 2020
>
> [2] Chen et al. Multi-scale spatial temporal graph convolutional network for skeleton-based action recognition. AAAI, 2021.

---

> > ### Author Response · Authors · 2024-12-02
> >
> > Thank you for your final review and recommendations. Regarding the final concerns you mentioned:
> >
> > > There should be some backbones besides multi-scale GCN and its variants [1][2] that can utilize multi-scale receptive fields or global receptive fields to learn action representations
> >
> > Masked Autoencoders are primarily designed for transformer-based architectures due to their ability to handle missing input tokens seamlessly. This approach is less applicable to Graph Convolutional Network (GCN) variants, as their operation fundamentally relies on the adjacency matrix to propagate information between nodes. Unlike transformers, where attention mechanisms enable flexible token interactions even with some tokens removed, GCNs depend on the complete graph structure. Removing a node disrupts the graph's connectivity and hampers knowledge transfer, making the masked autoencoder approach unsuitable for GCNs. As a result, it's not feasible to test on the mentioned GCN variants.
> >
> > > I still think a more comprehensive evaluation should be conducted, including more action recognition benchmark datasets such as NW-UCLA and more up-to-date MAE-based approaches
> >
> > Including the NW-UCLA dataset for additional evaluation could further strengthen our claim. However, we have already conducted evaluations on three widely used benchmarks featured in recent studies. Our experiments mirror the evaluation setup of MAMP, demonstrating that STARS effectively serves as a refinement module to achieve state-of-the-art performance across most metrics. Regarding newer MAE-based approaches, to our knowledge, MAMP represents the latest work in this category. Other subsequent methods either rely on contrastive learning from scratch or integrate MAE with contrastive learning. We compared against these and showed that STARS, with minimal contrastive tuning, consistently outperforms them.

---

### Official Review · Reviewer_qu9p · 2024-10-26

**Soundness:** 3
**Presentation:** 3
**Contribution:** 2
**Rating:** 5
**Confidence:** 5

**Summary:**

This paper proposes a new method for self-supervised skeleton action recognition. Specifically, it first pre-trains the encoder through the mask reconstruction in MAE, then a tuning strategy with contrastive learning to enhance the inter-class separability among actions. Extensive experiments demonstrate the superior performance of their method. Meanwhile, the code is released.

**Strengths:**

1. The paper is well-written, and the techniques sound reliable.
2. The work provides comprehensive experiments that are effective.

**Weaknesses:**

1. The novelty of this paper is limited. It seems like the composition of existing methods and the contribution is not clear.
2. Multi-stages pertaining is more complex than previous studies. Although the training time is decreased, the computation overhead must be considered.

**Questions:**

1. What is the difference between the core idea in this paper and the MAE-CT [1]? It seems that MAE-CT also trains MAE first and tunes the projector and predictor second. Besides, the pretraining pipeline utilized in this paper is MAMP, and the tuning framework used is the NNCLR. So, what is the novelty or the actual contribution of this work? Just employing them in the skeleton action recognition task? It is just an engineering work composed of several existing techniques.
2. Some state-of-the-art methods are not compared, e.g., PCM3 [2], UmURL [3]. The author should supply them for a comprehensive comparison.
3. Compared to the previous studies, the training time is decreased. However, the memory requirements look like they are increasing; 4 A40 GPUs are needed. Meanwhile, the queue size can influence this method's computation overhead. The author should supply the overhead comparison or other metrics to demonstrate the effectiveness of their method.

[1] Lehner J, Alkin B, Fürst A, et al. Contrastive tuning: A little help to make masked autoencoders forget[C]//Proceedings of the AAAI Conference on Artificial Intelligence. 2024, 38(4): 2965-2973.

[2] Zhang J, Lin L, Liu J. Prompted contrast with masked motion modeling: Towards versatile 3d action representation learning[C]//Proceedings of the 31st ACM International Conference on Multimedia. 2023: 7175-7183.

[3] Sun S, Liu D, Dong J, et al. Unified multi-modal unsupervised representation learning for skeleton-based action understanding[C]//Proceedings of the 31st ACM International Conference on Multimedia. 2023: 2973-2984.

---

> ### Author Response · Authors · 2024-11-22
>
> > What is the difference between the core idea in this paper and the MAE-CT [1]?
>
> Similar to MAE-CT, our approach is also a sequential approach that uses contrastive learning after MAE with the following differences:
>
> - The MAE-CT model is trained in three stages. In the second stage, the encoder is frozen, and only the NNCLR head is fine-tuned. In the third stage, both the backbone and NNCLR head are trained together. One motivation for this staged approach is the idea that the NNCLR head’s random weights could interfere with the representation quality by mapping the features into a random space, disrupting the learned structure. However, our findings challenge this assumption. The t-SNE visualization below demonstrates that even with random NNCLR head weights, the cluster structure in the encoder’s output space remains intact in the new representation space. Furthermore, we observed with STARS-3stage that replicating this three-stage process not only increases training time but also leads to a drop in final accuracy. Thank you for your question. We added the explanation in supplementary material. Please refer to Figure 6 in supplementary material to see the comparison.
> - MAE-CT applies data augmentation during pretraining, which works well for image data where transformations like color jittering or small rotations do not interfere with recognition. However, for action recognition, data augmentation can be risky, depending on the actions being distinguished in the dataset. For example, time flipping could make the model unable to distinguish between actions like standing up and sitting down, while spatial flipping could make it unable to tell the difference between raising the right hand and the left. To avoid these issues, STARS does not use any data augmentation.
>
> Another key contribution of this paper is highlighting a limitation in masked autoencoder methods like MAMP. While these methods often outperform contrastive learning approaches on many benchmarks, they fall short in few-shot settings, a scenario previously overlooked. We expose this gap and address it with a hybrid method that includes a few additional tuning steps during pretraining.
>
> > Some state-of-the-art methods are not compared, e.g., PCM3 [2], UmURL [3]. The author should supply them for a comprehensive comparison.
>
> Thank you for mentioning them. We included them in the revised manuscript.
>
> > Compared to the previous studies, the training time is decreased. However, the memory requirements look like they are increasing; 4 A40 GPUs are needed. Meanwhile, the queue size can influence this method's computation overhead. The author should supply the overhead comparison or other metrics to demonstrate the effectiveness of their method.
>
> Thank you for raising this concern. We used 4 A40 GPUs not because of the memory constraint but to leverage the PyTorch DDP for a faster pretraining in a limited time period as there are a lot of evaluations required to be done to prove the efficiency of the proposed method. Memory constraint as a computation overhead is a good concern that is not discussed in our submission . Here is the table we provided for the comparison, evaluated by measuring the memory consumed with a single input (Batch size = 1). Generally, MAMP requires more memory since it’s a heavy transformer-based approach. However they reduce this cost by passing 10% of tokens to the encoder and reconstruct the mask with the lightweight decoder. STARS use all the tokens and adds queues for contrastive learning, resulting in an increase in memory usage. AimCLR and ActCLR GCNs instead require much less memory. CMD uses three encoders in joint, motion, and bone at the same time and is GRU based, making it using the most memory usage. We also included this table in the revised supplementary material.
>
> | Method | Memory Usage  |
> |--------|--------------------|
> | MAMP   | 240 MB            |
> | AimCLR | 139 MB            |
> | ActCLR | 116 MB            |
> | CMD    | 1492 MB           |
> | STARS  | 998 MB            |

---

> > ### Comment · Reviewer_qu9p · 2024-11-25
> >
> > Thanks to the efforts of the authors. Most of my concerns have been addressed. However, I still feel that the contributions of this work to the community are limited, especially since the techniques it utilized have been explored and demonstrated in other tasks. Thus, I choose to maintain this score.

---

> > > ### Author Response · Authors · 2024-11-26
> > >
> > > Thank you for your detailed feedback and for acknowledging our efforts. While our work builds on established techniques, we believe it brings meaningful adaptations and insights specific to skeleton-based action recognition. These include avoiding augmentation pitfalls, improving efficiency by replacing the three-stage process in MAE-CT with a streamlined two-stage approach, and addressing the overlooked gap in few-shot scenarios with notable gains.
> > >
> > > Your suggestions, such as including comparisons with PCM3 and UmURL and providing a memory usage analysis, were particularly helpful in strengthening the manuscript and making it more comprehensive. We’re grateful for your thoughtful review, which has significantly improved our work. Thank you again for your time and valuable input!

---

### Official Review · Reviewer_iumL · 2024-10-27

**Soundness:** 3
**Presentation:** 2
**Contribution:** 2
**Rating:** 5
**Confidence:** 4

**Summary:**

This article proposes a human behavior analysis framework called STARS, aimed at improving the output representation of MAE encoders, thereby creating well-separated clusters without the need for any additional data augmentation. Extensive experiments and ablation studies conducted on three large-scale 3D skeleton action recognition datasets have verified the effectiveness of the STARS method.

**Strengths:**

1. This paper proposes the STARS framework, which combines MAE with contrastive learning, and can significantly improve the output representation of the MAE encoder with only a small amount of contrastive tuning.
2. Extensive experiments and ablation studies have been conducted on three large-scale 3D skeleton action recognition datasets, effectively proving the effectiveness of the method, and in most cases, reaching the state-of-the-art performance level.

**Weaknesses:**

1. Compared to some other contrastive learning methods (such as AimCLR, CMD), the STARS method only relies on single-view sequences for operation and does not use two different augmented views. Theoretically, its performance may be limited under cross-view evaluation on the NTU dataset. However, the cross-view evaluation experiment results in Table 1 and Table 3 are better than them. The article lacks relevant explanatory analysis.
2. As shown in the experimental results of Table 1, when the pre-trained and fine-tuned encoders are on a limited dataset, the difference in effect between the STARS method and other mask prediction methods (such as Skeleton MAE, MAMP) is not obvious, indicating that its generalization capability on small datasets needs further improvement.
3. The method uses some tricks in the ablation experiments, such as hierarchical learning rate decay (Formula 7); in addition, the Backbone is not aligned with other methods (Table 3), making it difficult to determine whether these tricks have brought benefits.
Experiments to be carried out:
4. Since the Backbone applied by the STARS method is different from other methods, its universality and effectiveness on different Backbones need to be verified (experiments need to be supplemented in Table 3).

**Questions:**

See the above weaknesses.

---

> ### Author Response · Authors · 2024-11-22
>
> > Compared to some other contrastive learning methods (such as AimCLR, CMD), the STARS method only relies on single-view sequences for operation and does not use two different augmented views....
>
> In Tables 1 and 3, STARS operates as a single-stream model, meaning it uses only joint coordinates as input throughout training and inference. In contrast, other models employ a three-stream approach. This involves: (1) training on joint coordinates, (2) training on bone information (the difference between adjacent joints like the wrist and elbow), and (3) training on velocity (the difference in joint positions between consecutive frames). At inference, these models take a skeleton sequence as input, compute the bone and velocity representations, and feed each into the three pretrained models. The outputs from these models are then combined using a weighted average, with weights chosen heuristically for optimal accuracy. STARS, however, simplifies this process by using a single encoder at inference that directly processes the joint coordinates, producing the final output without the need for multiple models or averaged outputs.
>
> However, in cross-view evaluation, the term "view" has a distinct meaning. It indicates that the same subject has repeated the same action in another recording from a camera positioned differently within the room. This setup presents challenges for video-based action recognition methods, which are sensitive to changes in viewpoint. For skeleton-based action recognition, however, all approaches apply a preprocessing step called canonicalization, aligning the torso with one of the XYZ axes in the first frame. This alignment minimizes the impact of viewpoint changes, so the model output remains consistent, and all models generally achieve higher scores in cross-view evaluations compared to cross-subject evaluations
>
> > As shown in the experimental results of Table 1, when the pre-trained and fine-tuned encoders are on a limited dataset, the difference in effect between the STARS method and other mask prediction methods (such as Skeleton MAE, MAMP) is not obvious, indicating that its generalization capability on small datasets needs further improvement.
>
> In the STARS framework, contrastive learning is used to enhance model generalization by pulling the representations of an anchor closer to its corresponding positive sample to reduce the distance while pushing other samples away. When the dataset size is limited, there are fewer points (sequence samples) in the model's representational space, making it more likely for the model to select a sample from a different action as the closest sample and applying contrastive learning. This can negatively affect the model’s performance, preventing it from improving much in these cases. Although leveraging approaches like data augmentation could help improve performance, we chose not to use it, as it can negatively impact some scenarios (see the first answer to reviewer zBeL for further details). As a result, the proposed STARS refinement method is most effective when the dataset size is not limited.
>
> > The method uses some tricks in the ablation experiments, such as hierarchical learning rate decay (Formula 7); in addition, the Backbone is not aligned with other methods (Table 3), making it difficult to determine whether these tricks have brought benefits.
>
> The idea behind hierarchical learning rate decay comes from the partial tuning approach described in the MAE paper (see Figure 9 in [3]). The MAE findings show that the early layers of a vision transformer already learn robust, general features during pretraining. When fine-tuning with labeled data, adjusting these early layers does not add much benefit. To make the most of the pretrained knowledge in these layers, we applied hierarchical learning rate decay, which prioritizes tuning the final layers. This reduces the risk of overwriting the foundational features learned in the initial training stage. Thank you for pointing this out; we will be sure to clarify this in the final manuscript.
> The choice of backbone is essential to the success of our method. Without the second training stage, the initial stage alone (the MAMP baseline) already outperforms other methods that use a different backbone. Our approach builds on this by introducing a refinement stage with a few additional fine-tuning epochs to further enhance MAMP’s performance. We also found that while MAMP exceeds the performance of other backbones, it falls short in few-shot scenarios that actions are not seen in pretraining (see Table 5 in the paper). Our refinement approach combines the strengths of both methods, leading to a substantial improvement in few-shot performance.
>
> [3] He, K., Chen, X., Xie, S., Li, Y., Dollár, P., & Girshick, R. (2022). Masked autoencoders are scalable vision learners. In Proceedings of the IEEE/CVF conference on computer vision and pattern recognition (pp. 16000-16009).

---

> > ### Author Response · Authors · 2024-11-22
> >
> > > Since the Backbone applied by the STARS method is different from other methods, its universality and effectiveness on different Backbones need to be verified (experiments need to be supplemented in Table 3).
> >
> > STARS leverages a combination of Masked Autoencoder (MAE) and Contrastive Learning (CL) in its pretraining stage. MAE is designed specifically for Transformer backbones, as only transformers have a global receptive field, allowing them to capture information from all tokens to accurately reconstruct missing content. With backbones lacking this global receptive field, MAE would likely fail to perform effectively, as they cannot leverage the full set of tokens needed for high-quality reconstruction.

---

> ### Author Response · Authors · 2024-12-01
>
> Dear Reviewer iumL,
>
> Thank you once again for your valuable time and thoughtful feedback on our paper! As the discussion period comes to an end, we would greatly appreciate your thoughts on our responses. If there are any remaining concerns or unclear points, we’d be happy to address them further.
>
> Best regards,
>
> The Authors

---

### Official Review · Reviewer_zBeL · 2024-11-02

**Soundness:** 3
**Presentation:** 3
**Contribution:** 2
**Rating:** 6
**Confidence:** 5

**Summary:**

- The paper tackles the problem of self-supervised learning for skeleton-representation learning.
- While previous state-of-the-art are MAE-based and achieve good fine-tuning performance, these suffer on linear evaluation, few shot learning due to poorly separated clusters.
- The paper proposes to have a two stage pre-training of MAE followed by a contrastive learning based training.
- The approach retains fine-tuning performance while being better at linear evaluation, and few-shot learning.
- The approach shows that a short contrastive learning stage is enough to obtain good performance.

**Strengths:**

- The approach is interesting and tries to leverage advantage of both masked-auto encoder and contrastive learning based pre-training.
- Simplicity of the approach is a strength of the paper. Approach needs minimal additional training to achieve the improvements.
- The paper is well written and easy to follow for the most part.
- The presented ablations and design decisions could be helpful to the community.

**Weaknesses:**

- Prior work: While simplicity is the strength for this paper. It proposes a combination of two existing approaches. The authors must make it clear if the two stages of training have any differences from the original approaches. A missing reference which also discusses differences in representations of MAE and CL-based pre-training for images and simple ways to use both [a]. Why was the proposed approach used instead of adapting one of the approaches in Section 2.2 for skeleton-based representation learning.
- How does training with NNCLR loss together with MAMP compare with the proposed approach ? What happens if the final stage employs both MAMP and NNCLR ?
- L231: "our method". Any changes from NNCLR must be clearly stated to make sure that the readers understand the differences.
- Experiments; Table 6 - why k=10 ? Table 2 uses k=1 making it difficult to compare. It would be interesting to have the same number of clusters (or both settings) to make comparisons easier. I am surprised that other approaches are so bad (Table 6) especially with MoCo which as the setup used in CMD.
- Since NNCLR was shown to be very effective post MAE training, do you have any baselines which uses NNCLR alone and compare it with MAMP and the proposed approach ?
- Table 5: How sensitive is the approach to different runs. How many times was this experiment repeated? Why do we see a drop from 1-shot to 2-shot ?


[a] What do self-supervised vision transformers learn?

Please also refer to section on "Questions"

**Questions:**

Please refer to the weaknesses section. Additionally:
- Do you use the same feature for linear evaluation, fine-tuning and kNN experiments ? Was one better over the other since a project was added ?

- Typo: L310 ViT?

---

> ### Author Response · Authors · 2024-11-22
>
> > Prior work: While simplicity is the strength for this paper. It proposes a combination of two existing approaches. The authors must make it clear if the two stages of training have any differences from the original approaches....
>
> In the first stage of training, we follow the exact approach used in MAMP, using the pretrained model checkpoints provided by the authors. However, our second stage diverges from traditional contrastive learning methods: Typically, contrastive learning relies heavily on data augmentation, which involves creating multiple augmented versions of a sequence as positive samples, with other samples in the batch or queue serving as negatives. While effective, this augmentation approach can inadvertently harm action recognition. For instance, certain augmentations, like time flipping, may be beneficial in some benchmarks but could hinder the model’s ability to differentiate between actions such as standing up versus sitting down, as it becomes invariant to temporal changes. Similarly, spatial flipping can reduce the model’s sensitivity to directional actions like raising the left hand versus the right. However, our contrastive tuning uses nearest-neighbor contrastive learning instead. Given that the MAMP-pretrained model already captures reliable features, we simply select the closest sample in the queue as the positive sample and bring it closer in the representational space, while pushing other samples away. This tuning step has three key benefits:
> 1) For actions where MAMP features are dispersed, tuning helps form well-defined clusters, as shown with the “hand-waving” example in Figure 3.
> 2) It eliminates the need for hand-crafted augmentations, making it adaptable across diverse datasets without the risk of augmentations that may harm performance (e.g., see the shearing and axis masking results in the table 7 of the paper).
> 3) It requires minimal additional training time, making it a practical refinement step for any MAE-based method, as illustrated in Figure 1.
> As for the reason why we rely on this sequential approach instead of similar hybrid approaches proposed in Section 2.2, it is mainly because of the high training costs associated with contrastive learning methods. Running these methods in parallel would require: 1) Data augmentation, because when training from scratch, relying on nearest samples as positive pairs is not effective—they are  randomly distributed and do not always yield good results. 2) A significant increase in training time. As Figure 1 illustrates, contrastive learning methods take much longer to train than MAMP. This is partly due to the need for multiple forward passes on each sequence, each with different augmentations for positive samples. In contrast, with the sequential approach, we only need to fine-tune the second stage with a minimal number of epochs, making it a lightweight and practical way to refine the learned representation.
>
> > How does training with NNCLR loss together with MAMP compare with the proposed approach ? What happens if the final stage employs both MAMP and NNCLR ?
>
> That is a good question. The table below shows the K-NN evaluation (K=1) of STARS vs the variant where the second stage uses both MAMP and NNCLR. STARS outperforms this approach. The table is also added into supplementary material of the paper.
> | Method         | NTU-60 XSub (%) | NTU-60 XView (%) |
> |----------------|----------------|-----------------|
> | MAMP           | 63.1           | 80.3            |
> | STARS          | 79.9           | 88.6            |
> | MAMP + NNCLR   | 74.6           | 86.5            |
>
> > Experiments; Table 6 - why k=10 ? Table 2 uses k=1 making it difficult to compare. It would be interesting to have the same number of clusters (or both settings) to make comparisons easier.
>
> We originally used k=10 since we used that to evaluate how effective our model is throughout the training, and used that for hyperparameter tuning. But later noticed that other methods used k=1 so we used that for comparison. Here we also provide the updated result for table 6 where k=1 is used.
>
> Updated table 6:
> | Tuning Strategy | NTU-60 XSub (%) | NTU-60 XView (%) |
> |------------------|-----------------|------------------|
> | DINO            | 69.1            | 84.0             |
> | DINO_aug        | 69.4            | 86.0             |
> | MoCo            | 63.4            | 85.7             |
> | MoCo_aug        | 64.3            | 88.5             |
> | NNCLR           | 79.9            | 88.6             |

---

> ### Author Response · Authors · 2024-11-22
>
> > Since NNCLR was shown to be very effective post MAE training, do you have any baselines which uses NNCLR alone and compare it with MAMP and the proposed approach?
>
> That is again a wonderful idea for an ablation study. In general, since the used NNCLR method does not rely on data augmentation and uses the sample with the least distance in the encoder’s representational space as the positive sample, it is tricky to tune an NNCLR from scratch since initially in the training because of the random weights the encoder has, the model maps the input sequence into a vector in output that nearest samples are not necessarily the samples with the same action. As a result, using the nearest samples as positive samples in contrastive loss is not a good heuristic. Table below shows that on K-NN evaluation, once we train the model from scratch using 300 epochs, it cannot distinguish the actions well. The table is also added into supplementary material of the paper.
>
> | k   | NTU-60 XSub (%) | NTU-60 XView (%) |
> |-----|-----------------|------------------|
> | 1   | 37.6            | 30.5            |
> | 2   | 35.8            | 28.7            |
> | 5   | 39.9            | 32.3            |
> | 10  | 41.2            | 33.6            |
>
> > Table 5: How sensitive is the approach to different runs. How many times was this experiment repeated? Why do we see a drop from 1-shot to 2-shot ?
>
> Following the approach in MotionBERT, we perform k-shot evaluation by measuring the similarity between each test sequence and all sequences in a set of unseen actions (i.e., actions that the model has not encountered during pretraining but for which we know the labels). Here is how it works: For each test sequence, we compute its distance from all other samples in the unseen set and select the k closest samples. Then, we assign the test sequence the label that appears most frequently among these k nearest neighbors (using majority voting). For example, let’s consider a 5-shot evaluation with three unseen actions, A, B, and C, each having 100 samples. For each test sample among these 300 samples, we compare it with the remaining 299 samples, identify the 5 nearest samples, and assign the label based on the majority action among these 5. In 2-shot evaluation, however, there can be cases where the two closest samples belong to different actions. When this happens, the test sample’s label is assigned randomly between the two, which can lead to a drop in accuracy. This accuracy drop is a useful indicator of how well-separated the clusters are in the representational space: a more substantial drop means that action clusters are less distinct, causing samples from different actions to be grouped together. Therefore, 2-shot evaluation is a good measure of the cluster separation quality for the 60 new unseen actions.
>
> > Do you use the same feature for linear evaluation, fine-tuning and kNN experiments?
>
> In both linear evaluation and k-NN experiments, we use the same features, which are extracted from the model backbone after pretraining without any access to actual labels. However, in fine-tuning, the features differ because the model backbone is further trained with supervised learning, where it sees the actual labels. This access to label information in fine-tuning provides a stronger signal for updating the backbone weights, resulting in improved features. As shown in Table 3, there is not a significant difference in performance between STARS and MAMP (the baseline). This suggests that while the proposed refinement technique in STARS enhances the representational space for unsupervised evaluations like linear probing and k-NN (where labels remain unseen by the backbone), the choice of STARS or MAMP as the starting point matters less in fully supervised fine-tuning. Nevertheless, both STARS and MAMP lead to much better results than starting with randomly initialized weights.
>
> > Typo: L310 ViT?
>
> That is a good point. Although skeleton sequences differ from visual perceptions (i.e. RGB images), we used the term ViT following the actual MAMP paper (See section 4.2 of [2]). That is because it follows the same design principle similar to Vanilla Vision Transformer. Instead of dividing the patch of images, now we have patches of joints in successive timeframes.
>
> > L231: "our method". Any changes from NNCLR must be clearly stated...
>
> Thank you for this suggestion. We will clarify this in the final version to ensure the differences are delineated.
>
> [1] Zhu, W., Ma, X., Liu, Z., Liu, L., Wu, W., & Wang, Y. (2023). MotionBERT: A unified perspective on learning human motion representations. In Proceedings of the IEEE/CVF International Conference on Computer Vision (pp. 15085-15099).
> [2] Mao, Y., Deng, J., Zhou, W., Fang, Y., Ouyang, W., & Li, H. (2023). Masked motion predictors are strong 3d action representation learners. In Proceedings of the IEEE/CVF International Conference on Computer Vision (pp. 10181-10191).

---

> > ### Comment · Reviewer_zBeL · 2024-11-24
> > **Thanks for the responses**
> >
> > Thanks for taking time to provide detailed responses. They were very helpful.
> >
> > - On prior work, two stage training: Some of the intuitions presented in the response to the first question are similar to those presented in this paper from last year [a] (which has already been cited) but is relevant given the responses. It is a purely contrastive learning-based approach but has the flavor of augmentation agnostic training using nearest points. Some discussion and contrast will be helpful to the reader. In-fact, the NNCLR-alone (w/ augmentations) would likely be very close to [a]
> >
> > In my opinion, transfer learning (w/ linear evaluation or fine-tuning) is often the most important use-cases for models pre-trained with SSL on large datasets. The presented approaches has mixed results there but particular shines when evaluating on the same dataset with linear evaluation protocol or kNN which is quite interesting too especially given the approach is simple. I am moving the score from 5 -> 6.
> > [a] HaLP: Hallucinating Latent Positives for Skeleton-based Self-Supervised Learning of Actions

---

> > > ### Author Response · Authors · 2024-11-26
> > >
> > > Thank you for your detailed review and insightful feedbacks

---

### Meta-Review · Area_Chair_WR6C · 2024-12-20

**Metareview:**

This paper introduces a human behaviour analysis method, STARS, aiming to enhance the output representation of MAE encoders. However, this work lacks sufficient novelty, and it seems to be a stack of existing techniques, like MAE and contrastive learning methods, to build the proposed method of 3D action recognition. Besides, multi-stages pertaining is more complex than previous works. The comparisons with other models are also vague and not specific. Due to the weaknesses, three reviewers recommend rejection.

**Additional Comments On Reviewer Discussion:**

Reviewers pointed out that this work lacks sufficient novelty, and it seems to be a stack of existing techniques, like MAE and contrastive learning methods, to build the proposed method of 3D action recognition. Besides, multi-stages pertaining is more complex than previous works. The comparisons with other models are also vague and not specific. The rebuttal addressed some of the concerns, yet most of reviewers are still not satisfied.

---

### Decision · Program_Chairs · 2025-01-22

Reject